# Personalized Healthcare for Dementia

**DOI:** 10.3390/healthcare9020128

**Published:** 2021-01-28

**Authors:** Seunghyeon Lee, Eun-Jeong Cho, Hyo-Bum Kwak

**Affiliations:** 1Program in Biomedical Science and Engineering, Inha University, Incheon 22212, Korea; slee@inha.edu (S.L.); cejeong97@naver.com (E.-J.C.); 2Department of Chemical Engineering, Inha University, Incheon 22212, Korea

**Keywords:** dementia, healthcare, diagnosis, engineering, exercise, diet

## Abstract

Dementia is one of the most common health problems affecting older adults, and the population with dementia is growing. Dementia refers to a comprehensive syndrome rather than a specific disease and is characterized by the loss of cognitive abilities. Many factors are related to dementia, such as aging, genetic profile, systemic vascular disease, unhealthy diet, and physical inactivity. As the causes and types of dementia are diverse, personalized healthcare is required. In this review, we first summarize various diagnostic approaches associated with dementia. Particularly, clinical diagnosis methods, biomarkers, neuroimaging, and digital biomarkers based on advances in data science and wearable devices are comprehensively reviewed. We then discuss three effective approaches to treating dementia, including engineering design, exercise, and diet. In the engineering design section, recent advances in monitoring and drug delivery systems for dementia are introduced. Additionally, we describe the effects of exercise on the treatment of dementia, especially focusing on the effects of aerobic and resistance training on cognitive function, and the effects of diets such as the Mediterranean diet and ketogenic diet on dementia.

## 1. Introduction

Dementia is the most common health problem in older adults. Worldwide, 47 million people have dementia, and this is expected to increase to 131 million by 2050 [1]. As the size of the population increases, the number of older people increases; accordingly, the population with dementia also increases [2]. It has been reported that 63% of the population will be affected by dementia in 2030, rising to 71% in 2050 [2]. The global cost of dementia in 2015 was estimated to be USD 818 billion, and this figure will increase as the number of people with dementia rises [3]. Dementia is a clinical syndrome rather than a specific disease. It is associated with a loss of cognitive abilities caused by irreversible brain disease or injury [2]. The most prevalent forms of dementia are Alzheimer’s disease (AD), vascular dementia (VD), and Lewy body dementia (LD). Risk factors for dementia include advanced age, genetic profile, systemic vascular disease, unhealthy diet, diabetes, hypertension, and physical inactivity [4]. Dementia is usually divided into two categories of disease: neurodegenerative and non-neurodegenerative [5]. Neurodegenerative dementia was originally referred to as “irreversible,” and non-neurodegenerative dementia was called “reversible”. Most cases of dementia in the aging population are caused by neurodegenerative diseases such as AD. AD is a gradual progressive loss of memory, especially memory of autobiographical information [6]. This is because AD affects brain networks responsible for episodic memory.

Parkinson’s disease dementia (PDD), which usually progresses from Parkinson’s disease (PD), is also a neurodegenerative disease. Similar to LD, Lewy bodies are abnormally deposited in the brain with PDD [7]. Further, LD and PDD show similar clinical symptoms [8]. However, patients with PDD have fewer visual hallucinations [9] and delusions [7] and more tremors at rest compared to those with LD [10]. Frontotemporal dementia (FTD), accompanying frontal lobe atrophy, usually has an onset age of 45–65 years, whereas common dementia usually has an onset age of over 65 years [11]. FTD can cause personality changes and cognitive disturbances, such as decreased concentration and speech impairment, but not memory loss in early stages [11]. Advanced FTD often leads to memory impairment, with a different pattern from AD. Patients with FTD have preserved implicit visual and verbal priming but memory deficit in encoding/retrieval. Further, patients with AD exhibit faster forgetting rates and encoding impairment compared to patients with FTD [12].

Mild cognitive impairment (MCI) is commonly caused by non-neurodegenerative processes. MCI is defined as having the ability to maintain daily function, but with lower-than-normal results in cognitive neuropsychological testing [13]. The treatment of dementia can be divided into pharmacological and nonpharmacological approaches. Management of cognitive and accompanying symptoms should include both pharmacological approaches, such as drugs, and nonpharmacological approaches, such as physical activity, diet, and monitoring systems. Without improved treatment approaches, including pharmacological and nonpharmacological treatments, the incidence of adverse consequences will continue to increase in dementia [14]. In this review, we summarize the diagnosis and treatment methods of dementia, focusing on nonpharmacological approaches, such as engineering design-assisted treatments, exercise, and diet.

## 2. Diagnosis of Dementia

### 2.1. Clinical Diagnosis

The diagnosis and distinction between different types of dementia are considered most important because the complexity and inconsistency of symptoms make the early stages of dementia challenging to distinguish. Dementia can be clinically characterized by progressive memory loss and impairment of cognitive functions or pathologically diagnosed by neuropathogenic changes. To date, practical methods of diagnosing dementia have begun with clinical approaches. There are several signs and symptoms, possibly indicating the need for evaluating dementia [15]: (1) cognitive changes, (2) psychiatric symptoms, (3) personality changes, (4) problematic behaviors, and (5) changes in day-to-day functioning. According to previous retrospective reports, personality changes in dementia generally include increased neuroticism and decreased extroversion, openness, and conscientiousness [16,17].

Based on the symptoms, a diagnostic sequence for dementia was suggested by Santacruz and Swagerty [18]. Based on the sequence of several questions, dementia can be differentiated from MCI. To date, the accuracy of the clinical diagnosis of dementia differs with the disease type, with an average accuracy of 80–90%. However, the early stages of dementia have not been fully evaluated because the initial personality changes frequently occur earlier than the clinical diagnosis [19]. Additionally, differentiating a specific type of dementia from other types of dementia is difficult because VD and LD, which are the most common types of dementia after AD, often occur in combination with AD [20,21]. Thus, clinically characterizing LD with concomitant pathological AD changes is nearly impossible.

### 2.2. Biomarkers

Biomarkers for dementia can be a promising option for the pathological diagnosis of dementia in early stages because they help objectively evaluate pathological sequences and disease progression in dementia [22,23]. Many trials have been aimed at discovering new biomarkers for dementia. Extracellular amyloid-β (Aβ) deposition and intracellular formation are the most common. Aβ and neurofibrillary tangles can contribute to the early detection of dementia. However, due to the high cost and invasive nature of their measurement, their applications are limited to research. Additionally, dementia has several potential biomarkers, including α-synuclein [24] and microRNAs (miRNAs) [25].

α-synuclein in the cerebrospinal fluid (CSF) can be used to differentiate LD from AD and healthy subjects. In a previous study, in patients with LD, α-synuclein levels were increased by 81.8% and 90% compared to control and AD groups, respectively [26]. However, the sensitivity was only approximately 56.2% and 50% higher compared to that of the control and AD groups, respectively. The effect of α-synuclein as a biomarker for LD can further improve with the tau level. In a previous study, for patients with clinically diagnosed α-synuclein-related disorders, including PD, LD, and essential tremors, the tau/α-synuclein ratio exhibited a greater area under the curve (0.8776), indicating a greater clinical accuracy compared to single tau (0.7739) and α-synuclein (0.7192) [27].

MiRNAs in biofluids can be used as a biomarker for neurodegenerative disease. They are present in all human biofluids with varying distributions [25]. In a previous study, among all miRNAs discovered in biofluids in AD, multiple sclerosis, PD, and amyotrophic lateral sclerosis, 33 and 10 miRNAs significantly differed in expression between the patients with multiple sclerosis and AD when compared to healthy controls, respectively [25]. Further, 38 miRNAs can play roles as biomarkers by distinguishing a minimum of two diseases. However, due to their poor stability, long-term storage issues must be addressed. Moreover, miRNA levels are significantly affected by red blood cell levels; subsequently, the miRNA composition can also be influenced by sample handing, to a difference of more than three orders [28]. Therefore, stability issues should be addressed.

Other actively investigated biomarkers for AD include nerve growth factor precursor proteins [29,30,31], which in previous studies were elevated in patients with AD in spite of the negligible difference in the nerve growth factor level compared to that in control groups, and mutations in the presenilin-1 or presenilin-2 gene [32,33,34,35], which affect the productions of Aβ40 and Aβ42 [36].

One popular source of biomarkers is blood samples. Blood can be easily extracted from patients at a low cost and provides a lot of information on bodily conditions. However, many biomarkers for neurodegenerative diseases do not exist at sufficient levels in the blood. To obtain pathological information regarding the central nervous system (CNS), CSF can be a candidate source for biomarkers of neurodegenerative diseases. The CSF surrounds the brain, and it protects and maintains the brain’s condition both physically and biologically.

Biomarkers for dementia are the best indicators for pathological changes in the brain. The Aβ/tau/neurodegeneration (A/T/N) classification system, proposed by the National Institute on Aging and Alzheimer’s Association, provides meaningful information on cognitive decline and the risk of dementia [37]. The Aβ biomarker is characterized with positron emission tomography (PET) or CSF Aβ42. The tau biomarker is characterized with tau PET or CSF p-tau. Neurodegeneration is characterized with CSF t-tau, fluorodeoxyglucose PET, or structural magnetic resonance imaging (MRI). “A”, “T”, and “N” are marked as positive or negative; for example, patients can be labeled A+/T-/N+. However, the results from the two characterizations are not always the same. For instance, when the results of PET and CSF Aβ differ, the patient might be labeled Ac.

Many studies have applied the A/T/N classification [38,39,40,41,42,43,44,45]. In one study, none of the patients with MCI labeled as A- progressed to AD, whereas several patients with MCI labeled as A+ progressed to AD [41]. In another recent study, which used the A/T/N biomarker classification system, the biomarkers were effective for screening patients with subjective cognitive decline, showing normal biomarkers, who were at the risk of clinical progression [45]. Indeed, the +/− categorization and the cutoff for a certain biomarker are limited in determining the pathological state [46], because the negative label does not always mean that the individual does not have brain pathology. However, current readouts for biomarkers are not sufficiently sensitive for low but important levels of biomarkers, suggesting the limits of biomarkers to reflect brain pathologies. Therefore, a biomarker classification system should include all possible biomarker profiles to be an effective tool [47,48].

### 2.3. Neuroimaging

Neuroimaging techniques that measure glucose metabolism, amyloid deposition, and the hippocampal volume on MRI are other approaches to diagnose AD [49,50]. Neuroimaging techniques are widely utilized to monitor white matter signal changes, spongiform and gliotic changes, and vascular damage. Furthermore, the pattern of atrophy, which refers to regional brain tissue loss, involves neuronal loss and is regarded as an indicator for a different type of dementia [51]. Serial MRI over time, ranging from several months to years, is commonly used as a safe method of monitoring neurodegeneration [52,53].

Neuroimaging can be structural, which includes computed tomography (CT) and MRI, or functional. AD, the most common type of dementia, can be differentiated by structural imaging based on brain atrophy in the medial temporal lobe (MTL). The MTL is present between multiple regions of the brain; thus, it is the interactive hub. It coordinates both learning and retrieval from the neocortex. Further, the hippocampus in MTL is related to spatial localization and associated with olfaction [54]. Thus, loss of smell frequently precedes dementia onset, and nasal administration of drugs can effectively treat dementia by providing a short path to MTL [55].

The sensitivity and specificity of distinguishing AD from VD and LD based on MTL atrophy can reach up to 91%, depending on the correlations between pathological investigation and MTL atrophy through Spearman’s rho [56]. The Braak stage (*p* = 0.022), which distinguishes between PD and AD, exhibits a significantly better performance in predicting MTL atrophy than the quantitative analysis of plaques (*p* = 0.375), tangles (*p* = 0.330), or Lewy bodies (*p* = 0.086). Additionally, the development of AD from MCI can be predicted by MTL atrophy with 73% sensitivity and 81% specificity [57,58]. Particularly, as a sensitive marker of progression of neurodegenerative disease, rates of either whole brain or hippocampal atrophy calculated from serial volumetric MRI have become increasingly used.

Functional imaging, however, reveals specific physiological activities. Functional imaging includes PET and single-photon emission tomography (SPECT), and typically utilizes probes or tracers to map the spatial distribution of the target. In PET, high cost cyclotron is required to generate radioisotopes with short half-lives. Among radioisotopes, ^11^C, ^15^O, and ^18^F are commonly used in PET. The metabolic tracer [^18^F]-2-fluoro-2-deoxy-D-glucose, which is a glucose analog, is widely used in PET examination for brain disorders [59]. The healthy brain consumes a large amount of glucose and subsequently exhibits active glucose metabolism, whereas the disordered brain shows impaired glucose metabolism. SPECT utilizes heavy radioisotopes with longer half-lives; thus, it is low-cost compared to PET. However, the collimator used in the γ camera to detect γ photons significantly lowers the sensitivity of SPECT; thus, SPECT exhibits lower sensitivity, compared to PET, of several orders [59]. The images that can be obtained from PET and SPECT depend on the probe. Probe compounds with small radioisotopes are used for PET. However, probe compounds with heavy radioisotopes, such as ^123^I [60,61] and ^99m^Tc [62,63], can be used for SPECT.

Several biomarkers, such as Aβ and tau, can be used in the functional imaging of dementia [64]. Furthermore, many features, including anatomy, connectivity, function, and vascular pathology, can be analyzed using functional imaging techniques (Figure 1) [65].

### 2.4. Digital Biomarkers

According to the Pew Research Center, the smartphone penetration rate is 95% in South Korea and 50% worldwide [66]. Additionally, the wearable device market, which includes smartwatches, is expanding rapidly. The dissemination of wearable devices and smartphones can be very useful in analyzing human behavioral patterns or even detecting physiological signals. Smartphones are equipped with approximately 15 types of sensors, such as a camera, microphone, touchscreen, and gyroscope. The structure and function of wearable devices differ depending on their purpose. Despite various purposes and functions, wearable devices generally have a common point of close contact with the human body for a long time. Thus, active data collection and passive data collection can be performed. These devices can extract many biological signals using a single sensor. For example, a gyroscope can detect walking, running, and cycling separately. However, combinations of several sensors can extract more complex and meaningful data from many biological signals.

Certain biological signals provide hints for dementia. For example, some behavioral patterns, such as sleep disturbances, agitation, and verbal aggression, are used to diagnose dementia clinically [67]. Such behaviors can be recognized through a combination of several biological signals obtained using smartphones and wearable devices [68]. There are several examples of sensors and biological signals that can be collected using wearable devices (Figure 2).

Microphones can sense vocal changes and detect the sleep pattern from the ambient noise generated during sleep. Fine motor controls, such as finger movement, can be recognized by touch screens and handwriting. Wandering outside as well as inside a house is a serious problem in dementia, and a global positioning system (GPS) can record the timeline of locations where a user wanders. Additionally, some physiological biosignals can provide more powerful information in a direct way. For example, electromyography (EMG), electrocardiography (ECG), and the chemical composition of blood already provide a lot of information regarding dementia.

Although the use of these digital biomarkers has not become widespread yet due to some drawbacks concerning medical information security, technical diagnosis accuracy, and the data processing capacity that can be achieved with a simple device, these challenges may be solved in the near future as the wearable device markets for personalized healthcare are rapidly growing. Moreover, wearable devices with advanced sensors will collect more valuable and complex information regarding the biomarkers that were previously described.

To date, although there are several diagnostic methods for dementia, such as clinical approaches, neuroimaging techniques, measurement of CSF levels of Aβ and tau (for AD), and digital biomarkers, both the sensitivity and specificity of diagnosis remains insufficient. Currently, characterization techniques using multiple markers are recommended.

## 3. Treatments

In addition to the diagnosis of dementia, treatment can also be assisted by engineering technologies. Specifically, long-term monitoring of body conditions during treatment should be performed with personalized healthcare because all patients have different lifestyles, genomes, and pathological histories. Among many engineering designs that can support the treatment of dementia, the monitoring system and drug delivery system (DDS) are representative. In addition, exercise and diet play an important role in treating dementia as nonpharmacological approaches.

### 3.1. Engineering Design-Assisted Treatments

#### 3.1.1. Monitoring System

Monitoring the physical and physiological conditions of the body is considered essential for treating patients with dementia via personalized healthcare. Some physical conditions investigated for a clinical diagnosis of dementia, such as problematic behaviors and changes in day-to-day functioning, can be collected using a personal monitoring system to detect cognitive changes, psychiatric symptoms, and personality changes. These changes can be detected from behaviors including voice tone, daily routine, and foot traffic using microphones, cameras, and GPS. Physical biosignals, such as blood pressure and heart rate, allow us to be aware of the biological condition and psychological state more directly.

Saif et al. reported the feasibility of using a wrist-worn biosensor device for at-risk AD patients [69]. The wrist-worn biosensor device collected data regarding sleep cycle, heart rate variability (HRV), and activity measures. The collected data were processed using unsupervised machine learning, followed by additional statistical analyses. The results showed a significant positive relationship between HRV and Dimensional Change Card Sort scores. This study demonstrated the potential of using wearable devices to monitor dementia.

Saied et al. demonstrated a wearable system for monitoring neurodegenerative diseases using flexible silicone-textile sensors and flexible switching circuits [70]. Wearable systems can monitor physiological changes in the brain from neurodegenerative diseases, including brain atrophy and lateral ventricle enlargement. This wearable sensor can detect brain atrophy and lateral ventricle enlargement separately in a quick and convenient way using different operating frequencies. The flexible nature of both the sensors and switching circuit allows the device to be used on the patient’s head.

In addition to using physical biosignals, several liquids, such as sweat, saliva, tears, and extracellular fluid, have been targeted as sources of physiological information that can be analyzed using wearable devices in a noninvasive manner. For example, biomarker proteins, such as Aβ and tau, are found in saliva, and their use is suggested for AD diagnosis [71]. MiRNAs are found in all human biofluids [25], such as tears, and can be used as a biomarker for AD [72].

Some studies have discovered several new biomarkers for various types of dementia. However, this approach is still in its early stages, and further studies should be performed before its practical application. Thus, we suggest further biosensors that might be utilized for dementia in the future. Kim et al. developed a smart biosensor system integrated with contact lenses (Figure 3) [73]. Contact lenses have an advantage as biosensors because of their continuous contact with tear fluid. Graphene and Ag nanowire hybrid electrodes were patterned on both sides of a flexible silicone elastomer substrate (Ecoflex). The device was flexible and stretchable, and only a negligible change in resistance was detected after 10,000 cycles of the stretching test. This biosensor was utilized to detect the concentration of glucose and intraocular pressure. As wires connected to lenses was impractical, an antenna was also designed to produce a wireless system. From in vivo testing in rabbits, both glucose concentration and intraocular pressure, which can be detected using a resonant frequency, could be collected by bringing the probe into contact with the eyes.

Furthermore, many engineering technologies, such as neural interface technologies [74], electronic skin (e-skin) [75,76], implantable electronics [77], and electroceuticals [78], provide significant advantages for various biomedical applications, including the monitoring and treatment of dementia [79]. However, several issues, including long-term biocompatibility and safety issues with new drugs, remain challenges. To overcome these challenges, various approaches such as bio-derived electronic materials [80,81], electronic material [82,83], biohybrid systems [84], and biomimetic designs [85] have been proposed; however, more breakthroughs are still needed in material development for the widespread use of advanced sensors.

#### 3.1.2. Drug Delivery System

Drug delivery system (DDS) refers to the technologies that optimize drug treatment by minimizing side effects and maximizing efficacy and effects by delivering the required amount of drug to the target [86]. Ideal DDS aims to deliver the required amount of drugs to the target tissue with appropriate timing. Chemical drugs generally diffuse throughout the body and cause side effects when they are not delivered to the exact target. DDS can address this issue and improve both the efficiency and efficacy of the drug by defining the target and controlling the timing and dosage of the drug. From the simplest devices that simply alert patients when to take their pills to advanced DDSs, many concepts to assist patients with taking their medication have been suggested [87,88]. In this section, we introduce several examples of advanced DDSs that can be potentially applied in dementia.

Sung et al. reported a flexible microdevice for drug delivery with controlled administration on a curved organ surface (Figure 4) [89]. A device comprising freestanding gold membranes over a reservoir array was fabricated. The optimized design of the device exhibited a stable performance at the applied current density for reliable drug release. Additionally, the device was powered using a wireless power transfer system that operated stably. The device was flexible and implantable on the curved cerebral cortex and successfully delivered two different chemicals.

Kaushik et al. reported a cerium oxide (CeO_2_) nanoparticle-mediated DDS using a computational biology approach to transport drugs through the blood–brain barrier (BBB) into the CNS [90]. The α-synuclein activity, which is a major factor causing PD, was elucidated in the presence of inhibitor nanoparticles of CeO_2_. The α-synuclein activity with biocompatible metal nanoparticles was also used to compare the efficacy and inhibition of CeO_2_ nanoparticles. The fitting of CeO_2_ nanoparticles was optimized to the active site of α-synuclein with intimate contacts and interactions. The features of CeO_2_ nanoparticles that inhibit α-synuclein offer the potential for employment as nanodrugs to treat dementia resulting from PD.

Drugs can be effectively delivered to the brain from the nasal cavity. Kamei et al. discovered the effect of enhanced nose-to-brain insulin delivery on progressive memory loss [91]. As insulin is considered as a candidate drug to treat dementia, the intranasal coadministration of insulin with cell-penetrating peptides (CPPs) was studied to promote the direct transportation of insulin from the nasal cavity into the brain parenchyma. As a result, the progression of memory loss as assessed using the spatial learning test was slowed. However, while this method was effective in the early stages of memory impairment, it was not effective in aiding recovery from the severe cognitive dysfunction accompanying Aβ accumulation in the brain, but rather increased Aβ plaque deposition in the hippocampus. Similarly, many new drugs to treat dementia have been developed. Since drugs can be delivered effectively using different methods, optimized DDS designs are needed.

### 3.2. Exercise

Physical exercise is essential to protect against dementia because it positively affects cognitive and physical functions, and also improves cardiovascular function. The main mechanism by which exercise affects cognitive function is an increase in blood flow to the brain, providing the necessary nutrients. Accordingly, exercise should be considered an essential feature of nonpharmacological approaches for dementia. We briefly describe the effects of aerobic, resistance, and combined exercise training on dementia and/or cognition. In many previous studies, aerobic exercise, especially moderate-to-high intensity exercise, leads to potent improvements in cognitive function (Table 1). Furthermore, memory function and functional mobility are improved by long-term aerobic exercise training. Cancela et al. showed that long-term cycling improves cognitive function, memory function, and functional mobility [92].

There have been far fewer studies of resistance exercise training than of aerobic exercise training for dementia or MCI. The potential utility of resistance exercise training is related to cognitive improvement [104,105]. Fiatarone et al. reported that 6 months of progressive resistance exercise training improved global cognition, executive function, and verbal memory in patients with MCI [99]. Moreover, Holthoff et al. showed that 12 weeks of resistance exercise training improved not only cognitive function but also behavioral and motor functions in AD [101].

There are also studies of combined exercise regimens and dementia. Bossers et al. reported the effects of combined exercise and aerobic-only exercise regimens on cognitive and motor functions in patients with dementia [102]. In this study, the combination of aerobic and resistance exercise training was more effective than aerobic-only exercise training in improving global cognitive, visual/verbal memory, and executive function. Table 1 summarizes the effects of aerobic, resistance, and combined exercise training on dementia. As a result, exercise training leads to significant improvements in cognitive performance, including memory and executive function. However, further studies are needed to identify the most beneficial type, intensity, and duration of the exercise program for dementia.

### 3.3. Diet

Diet influences dementia risk as a nonpharmacological treatment via multiple mechanistic pathways. Dietary changes can improve cardiovascular risk factors, decrease inflammation, and counteract oxidative stress. Additionally, recent studies on diet have focused on its role in dementia and/or cognition. A recent study reported that reducing the intake of saturated fats (e.g., processed meats) and increasing the intake of vegetables, legumes, fruits, and whole grains was associated with the improvement and prevention of decline in cognitive function in AD [106]. Moreover, several studies have studied the association between cognitive function and diets such as the Mediterranean diet, ketogenic diet, dietary approaches to stop hypertension, and caloric restriction [107,108]. Therefore, in this review, we focus on the effects of dietary patterns, such as fruits and vegetable consumption, on dementia and/or cognitive function.

#### 3.3.1. Mediterranean Diet

The Mediterranean diet (MeDi) is one of the most widely described and studied dietary patterns [109]. The MeDi varies by region and country, however, in general, it is characterized by high levels of consumption of vegetables, fruits, legumes, beans, nuts, cereals, grains, fish, and unsaturated fats such as olive oil. The positive effects of MeDi on cognitive function are probably mediated by reduced inflammation and oxidative stress (Figure 5) [110]. The main components of the MeDi, such as fruits and vegetables, reduce the brain impairment induced by reactive oxygen species (ROS) by reducing lipid oxidation and improving antioxidant defenses. Many studies have been conducted on MeDi and cognitive functions, memory function, and many other neurological functions. Some longitudinal studies have found beneficial effects of MeDi in MCI patients [111] or MCI conversion to AD [112]. Other studies also found that higher adherence to the MeDi was related to a significantly lower risk of developing AD [113,114]. Moreover, higher adherence to the MeDi was associated with a significant improvement in episodic memory and global cognition [115]. However, some studies have reported no association between MeDi and cognitive functions. A study showed that there were no effects of a higher adherence to the MeDi on the risk of developing dementia [116]. There are several reasons for this finding, such as the study design, test type, and the average age of the study population. Nevertheless, the majority of studies suggest that MeDi improves cognitive function among the elderly.

#### 3.3.2. Ketogenic Diet

The ketogenic diet (KD) is a very low carbohydrate and high fat diet. The KD converts the system from glucose metabolism to the production of ketone bodies and metabolism of fatty acids [118]. Ketosis is a metabolic state characterized by elevated levels of ketone bodies instead of glucose to produce energy. Since the 1920s, ketosis has been known for its neuroprotective role. The ketotic state is induced by fasting; a low-carbohydrate dietary pattern, such as KD, can achieve a state similar to the fasting state. There are several mechanisms by which the KD is protective against cognitive impairment (Figure 5). Among them, insulin resistance is a major factor for the progression of cognitive impairment, and the KD improves insulin secretion by beta-cells and enhances insulin sensitivity [119]. The positive effects of the KD have been reported in adults, in which high adherence to the KD significantly improved in logical memory tests after 12 weeks for patients with mild-to-moderate AD [120]. Moreover, previous studies have reported that the KD improves cognitive abilities, such as memory function, in patients with MCI or previously diagnosed with MCI [121]. Furthermore, the modified KD is related to increased cerebral perfusion and improvement of memory performance in older adults at risk of AD [122].

## 4. Conclusions

Dementia is a common disease that will affect an increasing percentage of the world’s population over the next 50 years. In this review, we provide several methods to diagnose and distinguish different types of dementia, including clinical approaches, biomarkers, and neuroimaging techniques. Furthermore, we have provided an overview of the treatments for dementia focused on nonpharmacological approaches including engineering design-assisted treatments, exercise, and diet.

There remain several hurdles to be overcome before a personalized healthcare system can be applied to patients with dementia. For the diagnosis of each type of disease associated with dementia, biomarkers should be increasingly researched for improved diagnostic accuracy. Moreover, new types of biomarkers, such as digital biomarkers, can be used for constant monitoring of the patient condition. In addition, advances in engineering designs, including advanced materials and data science, are accelerating technologies for the diagnosis and treatment of dementia. However, further studies are required for dementia biomarkers to be valuable research tools. Exercise and diet treatments provide meaningful benefits in the management of dementia. Most previous studies for dementia were performed in animal models. Direct evidence regarding the impact of engineering technologies on dementia should be demonstrated in a clinical setting.

## Figures and Tables

**Figure 1 healthcare-09-00128-f001:**
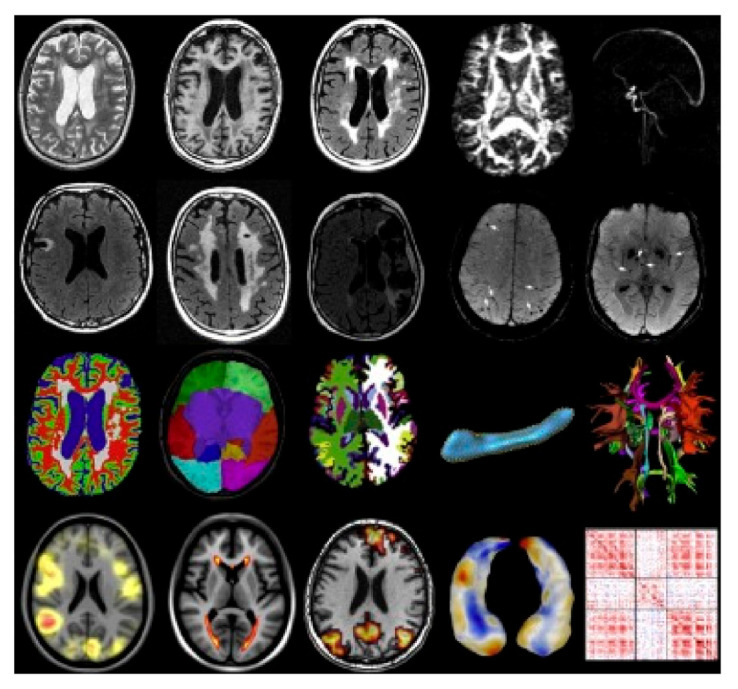
Examples of magnetic resonance imaging and quantitative imaging data of biomarkers in the brain. The biomarkers for quantitative imaging of the brain with dementia can be classified into four types: brain anatomy, brain function, and connectivity and vascular pathologies [65]. Reproduced with permission from Niessen, Medical Image Analysis; published by Elsevier, 2016.

**Figure 2 healthcare-09-00128-f002:**
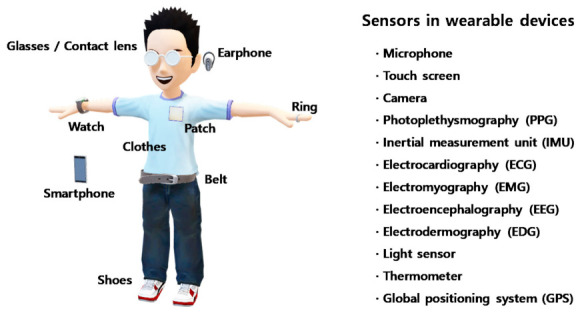
Various assistant devices and sensors that can be utilized for personalized healthcare. Microphones analyze changes in vocal and sleep patterns, and the touch screen detects the function of fine motor controls. The inertial measurement unit and electromyography sensor analyze the motor performance of the body. Photoplethysmography, electrocardiography, electrodermography, the light sensor, and the thermometer monitor the physiological condition, and electroencephalography directly senses the brain activity.

**Figure 3 healthcare-09-00128-f003:**
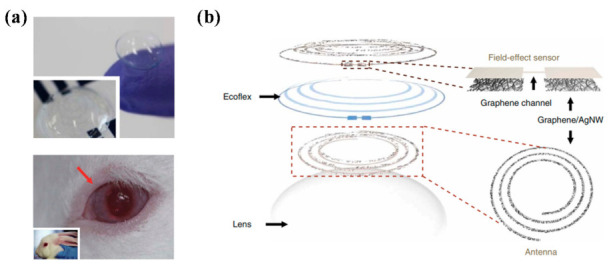
Wireless glucose and intraocular pressure sensor constructed on a contact lens. (**a**) Photographs of the transparent wireless contact lens biosensor (top) and the sensor applied to the rabbit’s eye (bottom); (**b**) configuration of the wireless contact lens biosensor [73]. Reproduced with permission from Kim, Nature Communications; published by Nature Publishing Group, 2017.

**Figure 4 healthcare-09-00128-f004:**
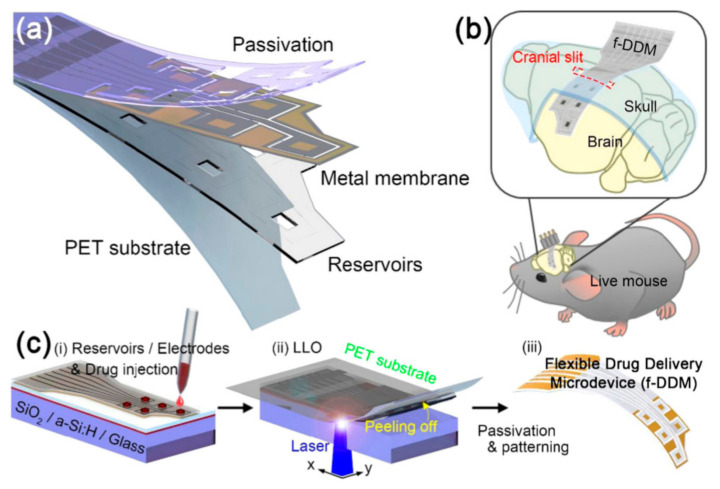
Structure and fabrication process of a flexible drug delivery microdevice [89]. (**a**) Components of the device composed of a positron emission tomography (PET) substrate, epoxy reservoir, metal membrane, and passivation; (**b**) schematic image of the device inserted in the skull of a live mouse; (**c**) fabrication process of the device. (i–iii) Drug injection, transfer process to the flexible PET, and fabricated device after the passivation process. Reproduced with permission from Sung, Nano Energy; published by Elsevier, 2018.

**Figure 5 healthcare-09-00128-f005:**
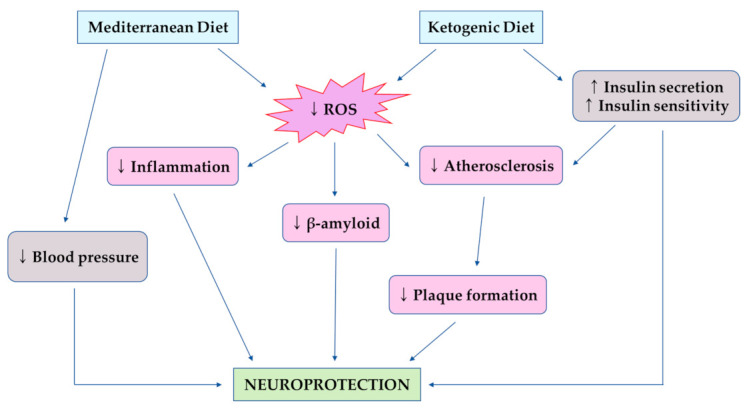
Protective effects of the Mediterranean diet and ketogenic diet on brain health and cognitive function [117]; ↑, increase; ↓, decrease.

**Table 1 healthcare-09-00128-t001:** Effects of aerobic, resistance, and combined exercise training on dementia.

Diagnosis	Age (yr)	Type of Exercise	Exercise Protocols	Results	References
Dementia	64–81.2	Aerobic	Treadmill, 30 min, 2 days, 16 weeks, moderate (60% of VO_2_ max) intensity	↑ Cognition	Arcoverde et al. (2014) [93]
MCI	55–80	Aerobic	40–60 min, 4 days, 6 months, high (75–85% maximum heart rate) intensity	↑ Executive function	Baker et al. (2010) [94]
Dementia	80.6 ± 8.3	Aerobic	Cycling, at least 15 min, 15 months	↑ Cognitive function ↑ Memory function ↑ Functional mobility	Cancela et al. (2016) [92]
AD	81.5 ± 6.6	Aerobic	Lower limb exercise 30 min, 5 days, 6 months, Low (40–60% maximum heart rate) intensity	↑ ADAS-cog	Kim et al. (2016) [95]
MCI	50–80	Aerobic	Cycling, 40 min, 3 days, 3 months, 70% of maximal intensity	↑ Cognitive function	Yang et al. (2015) [96]
MCI	74.5 ± 4.6	Aerobic	60 min, 3 days, 7 months	↑ Cognitive status	Maffei et al. (2017) [97]
AD	85	Aerobic	Walking, 60 min, 5 days, 12 weeks	↓ Sundowning syndrome	Venturelli et al. (2016) [98]
MCI	75	Resistance	100 min, 26 weeks	↑ Executive function ↑ Verbal/constructional memory ↑ ADAS-cog	Fiatarone et al. (2014) [99]
MCI	74–81	Resistance	15 repetitions, 12 weeks, 15RM (65% of 1RM)	↑ EEG patterns ↑ Cognitive function	Hong et al. (2018) [100]
AD	72.4 ± 4.3	Resistance	12 weeks	↑ Cognitive function ↑ Executive function ↑ Behavioral/motor function	Holthoff et al. (2015) [101]
Dementia	85.5 ± 5.1	Combined	2 days of resistance, 2 days of aerobic, 9 weeks, Moderate to high (50–85% maximum heart rate) intensity	↑ Global cognition ↑ Executive function ↑ Visual, verbal memory	Bossers et al. (2015) [102]
Dementia	81.8 ± 5.3	Combined	1 h, 3 days, 15 weeks	↑ Cognitive function ↑ Walking speed	Kemoun et al. (2010) [103]

MCI, mild cognitive impairment; AD, Alzheimer’s disease; ADAS-cog, Alzheimer’s Disease Assessment Scale; ↑, increase; ↓, decrease.

## Data Availability

No new data were created or analyzed in this study. Data sharing is not applicable to this article.

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
