# Peer review of "Personalized Healthcare for Dementia"

_healthcare, 2021, doi:10.3390/healthcare9020128_

Round 1

Reviewer 1 Report

With a few minor exceptions this manuscript very well written and it is a pleasure to read. The authors have an excellent command of written English. Despite this, or maybe because of this, the content of the paper in some of the subsections is disappointing from a scholarship perspective where it seems to lack enough detail to satisfy the interested reader. Other sections, specifically those describing assistive devices for mobility, seemed only remotely related to treating dementia and should be either expanded to justify inclusion or eliminated, thus preserving the overall length of the paper.

Overall, the paper is well-cited with many relevant references. One wishes that the details of the cited literature were actually discussed in the paper. What follows is in the spirit of improving a well-written paper on an important topic.

Specific comments follow.

Sections 1 and 2, Introduction and Diagnosis, are well written and nicely set the reader up for learning more about pharmacological and non-pharmacological treatments for the variety of dementias.

Line 82. “reach to 80-90%” is vague. What does this actually mean?

The paragraph on biomarkers, lines 92-104 is unsatisfactory from a scholarship perspective, i.e., it is devoid of sufficient detail. This is especially true since biomarkers hold the most promise for diagnosing dementias as they have the potential to be relatively inexpensive and easy to screen for in a clinical setting. For each one mentioned the following could be discussed: what is its role in the CNS? Which cells make it? What are microRNAs? What proteins do they encode? Why would any of these be increased in dementia? What are the data that support this biomarker’s use as a biomarker?

What are the pros and cons of each? For example, many cognitively unimpaired individuals die with brains full of tangles, tau and beta-amyloid. The same goes for brain atrophy.

The discussion of neuroimaging would benefit from some mention of why the memory encoding capacity of the medial temporal lobe/hippocampus makes these structures so important to the etiology of dementia. Also atrophy of the frontal lobe is thought to account for the affective components of dementia that often precede the memory issues. Again, a little more detail is needed. Why do we care about glucose metabolism?

Line 121. Please expand to clearly distinguish serial imaging over time (months? years?) verses serial sections obtained in a single MRI session, both of which are used in dementia diagnoses. (lines 132-134 seem to get at this issue in fact).

Line 126. How can atrophy be both “global” and specific to the medial temporal lobe?

Lines127-128. Please explain “early disproportionate symmetrical features” to the non-radiologist reader.

  1. As seem on an MRI, exactly what anatomical features differentiate MTL AD vs. LD vs. VD as verified by post-mortem inspection?

Lines 137-139. What’s the difference between PET and SPECT in terms of what can be imaged?

Figure 2. Lines141-143. Figure 2 is spectacular but the failure to describe it adequately in a caption or by referring to specific sub images this bit of text is unfortunate. “Various techniques” is woefully inadequate..

Lines 149-164. This change of topic to “wearable devices” occurs abruptly with no preamble or section title. Did something get deleted?

Line 165-166. “As mentioned above…” There does not seem to be a preceding paragraph that describes these behavioral changes. This would be a valuable addition to the paper, however.

Figure 3. Again, a very interesting image that needs a caption explaining what each device would measure and how it potentially could relate to diagnosing dementia. Lines 196-198 seem to suggest this technology is, in fact, not adequate to diagnosing dementia. As such it could probably be eliminated.

Monitoring devices section. Overall, this section would benefit from a careful rewrite that specifically describes what each device would measure and how the measured variable would relate to an individual’s cognitive state (e.g., lines 237-240).

Line 227. Do the authors really mean “treat” vs. “detect”?

Lines 265-273. This paragraph seems to discuss treatments, rather than monitoring.

Line 265. What are electroceuticals?

Lines 274-319. Exercise Assistance. In the absence of any real data showing these devices alter brain structure to ameliorate dementia, this section should be eliminated. In fact, it is hard to visualize an individual with severe cognitive impairment coping with these devices.

Line 299. Really? There are no motor axons in dorsal roots so it is hard to envision the neural pathway by which the stimulation would ultimately impinge on muscles.

Section DDS. Drug Delivery Systems? Please define. This section should be shortened to discuss only devices with direct impact on treating dementia. Parkinson’s related dementia needs to be introduced earlier in the paper.  Nasal administration of drugs would provide a short path to the mesial temporal lobe which is, in fact, the recipient of a large number of olfactory afferents. This is why loss of smell often precedes dementia onset (see R. Doty for a reference).

Lines 390-394. Eliminate as it reports negative results.

Table 1. One would hope that the arrow adjacent to the word “sundowning” should be pointing downward.

Mediterranean and Ketogenic diets sections. Very interesting and well written!

With a few minor exceptions this manuscript very well written and it is a pleasure to read. The authors have an excellent command of written English. Despite this, or maybe because of this, the content of the paper in some of the subsections is disappointing from a scholarship perspective where it seems to lack enough detail to satisfy the interested reader. Other sections, specifically those describing assistive devices for mobility, seemed only remotely related to treating dementia and should be either expanded to justify inclusion or eliminated, thus preserving the overall length of the paper.

Overall, the paper is well-cited with many relevant references. One wishes that the details of the cited literature were actually discussed in the paper. What follows is in the spirit of improving a well-written paper on an important topic.

Specific comments follow.

Sections 1 and 2, Introduction and Diagnosis, are well written and nicely set the reader up for learning more about pharmacological and non-pharmacological treatments for the variety of dementias.

Line 82. “reach to 80-90%” is vague. What does this actually mean?

The paragraph on biomarkers, lines 92-104 is unsatisfactory from a scholarship perspective, i.e., it is devoid of sufficient detail. This is especially true since biomarkers hold the most promise for diagnosing dementias as they have the potential to be relatively inexpensive and easy to screen for in a clinical setting. For each one mentioned the following could be discussed: what is its role in the CNS? Which cells make it? What are microRNAs? What proteins do they encode? Why would any of these be increased in dementia? What are the data that support this biomarker’s use as a biomarker?

What are the pros and cons of each? For example, many cognitively unimpaired individuals die with brains full of tangles, tau and beta-amyloid. The same goes for brain atrophy.

The discussion of neuroimaging would benefit from some mention of why the memory encoding capacity of the medial temporal lobe/hippocampus makes these structures so important to the etiology of dementia. Also atrophy of the frontal lobe is thought to account for the affective components of dementia that often precede the memory issues. Again, a little more detail is needed. Why do we care about glucose metabolism?

Line 121. Please expand to clearly distinguish serial imaging over time (months? years?) verses serial sections obtained in a single MRI session, both of which are used in dementia diagnoses. (lines 132-134 seem to get at this issue in fact).

Line 126. How can atrophy be both “global” and specific to the medial temporal lobe?

Lines127-128. Please explain “early disproportionate symmetrical features” to the non-radiologist reader.

  1. As seem on an MRI, exactly what anatomical features differentiate MTL AD vs. LD vs. VD as verified by post-mortem inspection?

Lines 137-139. What’s the difference between PET and SPECT in terms of what can be imaged?

Figure 2. Lines141-143. Figure 2 is spectacular but the failure to describe it adequately in a caption or by referring to specific sub images this bit of text is unfortunate. “Various techniques” is woefully inadequate..

Lines 149-164. This change of topic to “wearable devices” occurs abruptly with no preamble or section title. Did something get deleted?

Line 165-166. “As mentioned above…” There does not seem to be a preceding paragraph that describes these behavioral changes. This would be a valuable addition to the paper, however.

Figure 3. Again, a very interesting image that needs a caption explaining what each device would measure and how it potentially could relate to diagnosing dementia. Lines 196-198 seem to suggest this technology is, in fact, not adequate to diagnosing dementia. As such it could probably be eliminated.

Monitoring devices section. Overall, this section would benefit from a careful rewrite that specifically describes what each device would measure and how the measured variable would relate to an individual’s cognitive state (e.g., lines 237-240).

Line 227. Do the authors really mean “treat” vs. “detect”?

Lines 265-273. This paragraph seems to discuss treatments, rather than monitoring.

Line 265. What are electroceuticals?

Lines 274-319. Exercise Assistance. In the absence of any real data showing these devices alter brain structure to ameliorate dementia, this section should be eliminated. In fact, it is hard to visualize an individual with severe cognitive impairment coping with these devices.

Line 299. Really? There are no motor axons in dorsal roots so it is hard to envision the neural pathway by which the stimulation would ultimately impinge on muscles.

Section DDS. Drug Delivery Systems? Please define. This section should be shortened to discuss only devices with direct impact on treating dementia. Parkinson’s related dementia needs to be introduced earlier in the paper.  Nasal administration of drugs would provide a short path to the mesial temporal lobe which is, in fact, the recipient of a large number of olfactory afferents. This is why loss of smell often precedes dementia onset (see R. Doty for a reference).

Lines 390-394. Eliminate as it reports negative results.

Table 1. One would hope that the arrow adjacent to the word “sundowning” should be pointing downward.

Mediterranean and Ketogenic diets sections. Very interesting and well written!

With a few minor exceptions this manuscript very well written and it is a pleasure to read. The authors have an excellent command of written English. Despite this, or maybe because of this, the content of the paper in some of the subsections is disappointing from a scholarship perspective where it seems to lack enough detail to satisfy the interested reader. Other sections, specifically those describing assistive devices for mobility, seemed only remotely related to treating dementia and should be either expanded to justify inclusion or eliminated, thus preserving the overall length of the paper.

Overall, the paper is well-cited with many relevant references. One wishes that the details of the cited literature were actually discussed in the paper. What follows is in the spirit of improving a well-written paper on an important topic.

Specific comments follow.

Sections 1 and 2, Introduction and Diagnosis, are well written and nicely set the reader up for learning more about pharmacological and non-pharmacological treatments for the variety of dementias.

Line 82. “reach to 80-90%” is vague. What does this actually mean?

The paragraph on biomarkers, lines 92-104 is unsatisfactory from a scholarship perspective, i.e., it is devoid of sufficient detail. This is especially true since biomarkers hold the most promise for diagnosing dementias as they have the potential to be relatively inexpensive and easy to screen for in a clinical setting. For each one mentioned the following could be discussed: what is its role in the CNS? Which cells make it? What are microRNAs? What proteins do they encode? Why would any of these be increased in dementia? What are the data that support this biomarker’s use as a biomarker?

What are the pros and cons of each? For example, many cognitively unimpaired individuals die with brains full of tangles, tau and beta-amyloid. The same goes for brain atrophy.

The discussion of neuroimaging would benefit from some mention of why the memory encoding capacity of the medial temporal lobe/hippocampus makes these structures so important to the etiology of dementia. Also atrophy of the frontal lobe is thought to account for the affective components of dementia that often precede the memory issues. Again, a little more detail is needed. Why do we care about glucose metabolism?

Line 121. Please expand to clearly distinguish serial imaging over time (months? years?) verses serial sections obtained in a single MRI session, both of which are used in dementia diagnoses. (lines 132-134 seem to get at this issue in fact).

Line 126. How can atrophy be both “global” and specific to the medial temporal lobe?

Lines127-128. Please explain “early disproportionate symmetrical features” to the non-radiologist reader.

  1. As seem on an MRI, exactly what anatomical features differentiate MTL AD vs. LD vs. VD as verified by post-mortem inspection?

Lines 137-139. What’s the difference between PET and SPECT in terms of what can be imaged?

Figure 2. Lines141-143. Figure 2 is spectacular but the failure to describe it adequately in a caption or by referring to specific sub images this bit of text is unfortunate. “Various techniques” is woefully inadequate..

Lines 149-164. This change of topic to “wearable devices” occurs abruptly with no preamble or section title. Did something get deleted?

Line 165-166. “As mentioned above…” There does not seem to be a preceding paragraph that describes these behavioral changes. This would be a valuable addition to the paper, however.

Figure 3. Again, a very interesting image that needs a caption explaining what each device would measure and how it potentially could relate to diagnosing dementia. Lines 196-198 seem to suggest this technology is, in fact, not adequate to diagnosing dementia. As such it could probably be eliminated.

Monitoring devices section. Overall, this section would benefit from a careful rewrite that specifically describes what each device would measure and how the measured variable would relate to an individual’s cognitive state (e.g., lines 237-240).

Line 227. Do the authors really mean “treat” vs. “detect”?

Lines 265-273. This paragraph seems to discuss treatments, rather than monitoring.

Line 265. What are electroceuticals?

Lines 274-319. Exercise Assistance. In the absence of any real data showing these devices alter brain structure to ameliorate dementia, this section should be eliminated. In fact, it is hard to visualize an individual with severe cognitive impairment coping with these devices.

Line 299. Really? There are no motor axons in dorsal roots so it is hard to envision the neural pathway by which the stimulation would ultimately impinge on muscles.

Section DDS. Drug Delivery Systems? Please define. This section should be shortened to discuss only devices with direct impact on treating dementia. Parkinson’s related dementia needs to be introduced earlier in the paper.  Nasal administration of drugs would provide a short path to the mesial temporal lobe which is, in fact, the recipient of a large number of olfactory afferents. This is why loss of smell often precedes dementia onset (see R. Doty for a reference).

Lines 390-394. Eliminate as it reports negative results.

Table 1. One would hope that the arrow adjacent to the word “sundowning” should be pointing downward.

Mediterranean and Ketogenic diets sections. Very interesting and well written!

Reviewer 2 Report

Dear authors, 

Thanks for sending your manuscript to the Healthcare journal. This manuscript is a good contribution to dementia management. However, there are some comments you need to address. 

  1. You need to change the organization of your manuscript. For example, (1st) Introduction about dementia, (2nd) Diagnosis of dementia, (3rd) Pharmacological treatment, (4th) Non-Pharmacological treatment, and (5th) New technology for Personalized Healthcare dementia. 
  2. Some sentences and data in your manuscript are not well supported, please review some sentences without references. 
  3. Can you improve your figure 1? Please, it's difficult to read it and follow the line. 
  4. Please add more data about how the biomarker behaves in a clinical setting? Is there a cut-off point available? 
  5. Please explain the disadvantages of the majority of technology in Personalized Healthcare dementia. One of them is: the vast evidence is from animals. 

Reviewer 3 Report

In this manuscript, the authors discussed different diagnostic and treatment approaches for dementia with the aim of personalized healthcare. I personally found this review very interesting and original. It’s also well-written and organised. The figures are exhaustive and very clear. I have only few suggestions:

  • Please insert the ATN classification system.
  • I should suggest the authors to better explain possible implications of their manuscript and to extend the conclusions.

Round 2

Reviewer 1 Report

Congratulations to the authors on a fine revision. I very much enjoyed re-reading it- and learned a bit as well. It should be of interest to a wide audience. I hope you receive many citations!

Reviewer 2 Report

Dear authors, 

Thanks for addressing the comments in your previous version of your work. This manuscript is a good contribution to the current field of dementia and the technology used for treatment. 

In my opinion, this version could accept it.